# Using machine learning to predict judgments on Western visual art along content-representational and formal-perceptual attributes

**Blanca T. M. Spee**[1,2,3]*, **Helmut Leder**[1,3], **Jan Mikuni**[1], **Frank Scharnowski**[3],
**Matthew Pelowski**[1,3], **David Steyrl**[3]

1 Vienna Cognitive Science Hub, University of Vienna, Vienna, Austria, 2 Center of Expertise for Parkinson & Movement Disorders, Department of Neurology, Donders Institute for Brain, Cognition and Behavior, Radboud University Medical Center, Nijmegen, The Netherlands, 3 Department of Cognition, Emotion and Methods in Psychology, Faculty of Psychology, University of Vienna, Vienna, Austria

* blanca.spee@univie.ac.at

## Abstract

Art research has long aimed to unravel the complex associations between specific attributes, such as color, complexity, and emotional expressiveness, and art judgments, including beauty, creativity, and liking. However, the fundamental distinction between attributes as inherent characteristics or features of the artwork and judgments as subjective evaluations remains an exciting topic. This paper reviews the literature of the last half century, to identify key attributes, and employs machine learning, specifically Gradient Boosted Decision Trees (GBDT), to predict 13 art judgments along 17 attributes. Ratings from 78 art novice participants were collected for 54 Western artworks. Our GBDT models successfully predicted 13 judgments significantly. Notably, judged creativity and disturbing/irritating judgments showed the highest predictability, with the models explaining 31% and 32% of the variance, respectively. The attributes emotional expressiveness, valence, symbolism, as well as complexity emerged as consistent and significant contributors to the models' performance. Content-representational attributes played a more prominent role than formal-perceptual attributes. Moreover, we found in some cases non-linear relationships between attributes and judgments with sudden inclines or declines around medium levels of the rating scales. By uncovering these underlying patterns and dynamics in art judgment behavior, our research provides valuable insights to advance the understanding of aesthetic experiences considering visual art, inform cultural practices, and inspire future research in the field of art appreciation.

## Introduction

The way we assign value to a work of art is, to a great extent, a matter of judgments [1]. It begins with the question of whether a work qualifies as art, marking the first step in a

**Data Availability Statement:** The datasets generated and/or analyzed during the current study, along with detailed results on how each

predictor influences the prediction outcomes, are available in the GitHub repository at https://github.com/univiemops/art-rating-prediction. Artwork images used from the Vienna Art Picture System (VAPS, see for full list of stimulus-set rated including VAPS-identification code S3 Table) are not publicly available due to copyright restrictions concerning the artists. However, these can be requested at the Faculty of Psychology, University of Vienna, Vienna, Austria (dekanat.psychologie@univie.ac.at , see also publication on VAPS at https://doi.org/10.1037/aca0000460).

**Funding:** The author(s) received no specific funding for this work.

**Competing interests:** The authors have declared that no competing interests exist.

continuum of assessment [2]. Once a piece is recognized as art, individuals express their engagement through various judgments associated with visual art and aesthetic experiences [3, 4]. These judgments encompass concepts such as beauty, liking, interest, and thought-provoking, representing a means to articulate subjective evaluations [1, 3, 5]; evaluations that are deeply influenced by the individual history of encounters with art within social and cultural contexts [6–9].

Attributes, on the other hand, play a pivotal role in imbuing these judgments with subjective characteristics. Specifically, someone's preference for an artwork or the recognition of a masterpiece by certain cultural groups, laymen, or experts go beyond mere artist recognition or historical references [10, 11]. Attributes are specific qualities individuals employ to substantiate their evaluations, forming the foundation for articulating and validating subjective opinions. For instance, one might attribute their appreciation of a work of art to its colorfulness, find beauty in emotional expressiveness, associate imaginativeness with creativity [12], or experience fascination due to unique stylistic elements [3–5, 13, 14 see for further examples, 15]. These attributes serve as criteria for evaluating and categorizing artworks, including factors such as complexity, color usage, emotional expressiveness, stylistic elements, and other observable features. By considering these attributes, judgments acquire personalization and nuance, adding subjective qualities to the overall evaluation of artworks.

Our journey into understanding art judgments and their connection to art attributes began with a focused study on creativity judgments. In this prior work [12], we employed interpretable machine learning using Random Forest ensemble models [16] to probe the complex associations between 17 subjective art attributes and creativity judgments across a diverse range of artworks. While attributes like symbolism, emotional expressiveness, and imaginativeness played significant roles in predicting creativity judgments, other factors like abstractness, valence, and complexity also made an impact, albeit to a lesser degree. This investigation presented the first attribute-integrating quantitative model of factors contributing to creativity judgments in visual art among novice raters. It marked a significant stride forward in building the groundwork for introducing machine learning as an innovative approach, by focusing specifically on subjective attributes ratings predicting judgments.

Building upon this foundation, our current study extends the scope beyond creativity judgments to encompass a broader spectrum of 13 different art judgments. We used the same set of 17 art attributes [12]. Our approach remains rooted in machine learning, with the aim of unveiling intricate patterns, regularities, and potential inconsistencies within art evaluation. However, we updated our model, specifically applying Gradient Boosted Decision Trees [GBDT, 17]. By leveraging data-driven analyses, we seek to enhance our understanding of art judgment behavior and the appreciation of art, shedding light on the multifaceted interplay between judgments and attributes.

## The societal significance of understanding art judgments and attributes

Understanding art judgments and their connection to attributes is of profound importance in both contemporary society and the realm of art and aesthetic research [1, 18]. When people talk about art and their preferences, judgments, values, and explanatory attributes are inherently tied to the attempt to communicate, might it be recognizing some information about people themselves—such as social reputation or sparking intellectual discourse [7, 9, 10]—or about societal issues, fostering critique, provocation, and progress within societies [8, 19]. The connection between attributes and art judgment has also been used to test universal assumptions of art historical claims, which largely rely on the coherence of these connections [4]. In Specker and colleagues, for example, they explored the assumption of universality in the

perception of aesthetic effects associated with visual objective elements such as colors and lines. They investigated whether people universally associate certain qualities with key visual forms, a notion fundamental to art discourse. Their findings challenged this assumption, revealing significantly lower agreement than expected on several aesthetic effect dimensions. While participants agreed on the effects of warm-cold, heavy-light, and happy-sad, there was hardly any consensus on 11 other dimensions. Intriguingly, consensus was higher regarding the whole artwork, rather than for single elements within the artwork, potentially addressing the importance of more holistic content-representational attributes rather than objective, or formal perceptual features [18]. Notably, our prior research into creativity judgments revealed that certain attributes, particularly content-representational features, significantly predict creativity judgments, offering a deeper understanding of this intricate relationship [12]. This highlights the complexity of art judgments and the need to explore the role of attributes further.

Artistic creations possess the remarkable capacity to challenge established norms, ignite intellectual discourse, and inspire change. Consequently, understanding art judgments allows us to gain insights into how individuals perceive and respond to artistic interventions that seek to question, disrupt, or redefine societal narratives (Becker, 1982). Through this lens, we can comprehensively analyze the diverse range of responses and interpretations evoked by artworks, thereby contributing to a deeper comprehension of the cultural and social impact of art [20].

Furthermore, the relevance of studying art judgments is magnified in today's digital worlds [21]. The proliferation of online platforms and social media has democratized art consumption, making artworks more accessible than ever before. Platforms like Pinterest and Instagram have fundamentally transformed the way art is discovered, shared, and evaluated in digital spaces [22]. This heightened accessibility underscores the necessity of a comprehensive understanding of art judgments. By delving into the relationship between art judgments and attributes, we can gain profound insights into the mechanisms that underpin the appreciation and dissemination of art in the digital realm as well. Our research, building on our prior work [12], strives to uncover how attributes, particularly those linked to subjective ratings, influence the judgments placed on artworks, extending our knowledge in the field of art and creativity research.

## The study of art judgments and machine learning: A historical perspective and future prospects

The exploration of art judgments and their association with art attributes has a rich history that stretches back nearly a century, although philosophical discussions on the topic likely trace back to the first expressions of human creativity [23–25]. Among the early pioneers in this field, Daniel Berlyne stands out for introducing multidimensional scales and focusing on hedonic judgments and attributes related to complexity and arousal [5, 26–29]. However, Berlyne's approach, which included the measurement of "cortical arousal" (albeit in the form of ratings), faced criticism for its reductionistic view of aesthetic experience and often encountered challenges in replication [30–33]. These limitations led to further research in the field. Despite ongoing research on art judgments and art appreciation models [3], there remains a significant gap in the analysis of the predictability of factor clusters and the intricate relationship between art judgments and attributes [2, 34]. Current studies often focus on a limited number of commonly used ratings, such as liking, familiarity, understanding, and beauty, without thoroughly investigating the validity of these judgments or the attributes that underlie them [35, 36]. Additionally, there is often a conflation of art judgments and attributes, with attributes frequently not explicitly tested for their role in shaping art judgments [18].

Existing research on art judgments and their connection to art attributes faces two primary limitations. First, the field lacks a comprehensive and systematic analysis of the variables and dimensions involved in art judgments and their interrelationships. While there have been extensive studies on individual judgments and attributes, such as beauty, complexity, liking, and others, there is a pressing need to explore a broader spectrum of factors and how they interact in the evaluation of artworks. Understanding the complex dynamics of art judgments and the influence of attributes can provide valuable insights into the multifaceted nature of aesthetic experiences [37].

Second, the majority of studies in this area have traditionally relied on linear models for analysis [38]. These models assume linear relationships between judgments and attributes, which may not adequately capture the intricate and nonlinear nature of art evaluation as a human behavior. To address these limitations, our research embraces machine learning techniques and non-linear models to explore the complex and nuanced relationships between judgments and attributes [39]. Machine learning offers flexible and robust modeling approaches capable of handling high-dimensional data and capturing nonlinear patterns, allowing us to uncover hidden relationships and identify novel factors that influence art judgments [16, 40–44].

The integration of computational models into the realm of aesthetics began in the 1960s when Max Bense and G. D. Birkhoff laid the foundations for computational aesthetics and the quantification of aesthetics using mathematical formulas [45, 46]. Bense aimed to derive scalar measurements of artwork aesthetics, merging information theories and generative language theories. Intriguingly, Bense's approach was influenced by Berlyne's studies on attribute inter-relations for preference and aesthetics [5, 28, 47].

While machine learning has gained prominence in various fields exploring human behavior, psychology, and decision processes [16, 40–44], its application in art research has been relatively limited. Studies by Li and colleagues [48, 49] have used machine learning to investigate the predictability of subjective aesthetic preferences based on visually objective properties, such as formal-perceptual features, of artworks. Their findings suggest that objective artwork features can inform predictions of preference ratings, hinting at a degree of universality in the evaluation of visual art objects, at least on a perceptual level (although note the discussions presented by Specker et al., 2020).

Research conducted by Iigaya and colleagues [50–52] has delved into the neural mechanisms underpinning preference computations in the brain. They have revealed that preference computations involve a graduated hierarchical representation of attribute structures in the visual system, with attribute information influencing subjective evaluation in higher cognitive brain areas. This aligns with the argument that information processing in art judgments may involve shared brain networks [35], hereby supporting Berlyne's starting viewpoint on art psychobiology [26] and subsequent work in neuroaesthetics [53–55]. Iigaya's and colleagues research indicates interpersonally shared attribute-specific processing, aligning with the notion that judgments rely on factors patterned along socially and culturally learned attributes [6, 56, 57].

In summary, all these findings encourage the use of machine learning to model art judgments using attributes as features derived from artworks. Leveraging machine learning algorithms allows to construct data-driven models capable of capturing the nuanced dynamics of art evaluation. Notably, our prior research on that matter [12] shows that using non-linear machine learning models is most informative also for subjective ratings.

## Methods literature review: Procedure and identification of factors

A systematic literature review was conducted, in line with the guidelines for systematic reviews, to identify potential attributes and art judgments [58]. Based on the review, we developed two sets of scales (1) art judgments and (2) art attributes. These scales were used in the empirical study for the ratings of the artworks. Additionally, the summary of publications (S1 Appendix) might itself serve as a comprehensive summary of the current state of research in the field and inspire future studies, such as determining which measures should be used for specific purposes and identifying aspects that have yet to be investigated.

## Identification of art judgments and art attributes

Results of the systematic review adhered to established guidelines for systematic reviews [58], and the process is detailed in S1 Fig. We conducted a thorough search using various electronic databases and search engines, including Google Scholar, PEDro, Scopus, and PubMed, in addition to academic libraries accessible to University of Vienna staff. Initially, we employed keywords such as "art," "art judgment/judgement," "art evaluation," "art scales," "art assessment," "artistic," and "aesthetic." In a subsequent search, we included sub-keywords such as "attributes," "features," "dimensions," "criteria," "factor," "factor-analyses," "machine learning," and "statistic learning approach." For each keyword, we examined the first five pages of results, yielding 76 full-text publications. After applying relevance criteria related to art judgments and attributes, 46 publications were selected for further analysis. By scrutinizing the references within these publications, we identified an additional 14 relevant publications. In total, our review encompassed 33 art assessment publications and three studies at the intersection of art and machine learning. The development of two sets of scales, one for art judgments and one for art attributes, drew from a comprehensive pool of 36 publications. For an overview of the approaches and reviews, including relevant papers not explicitly discussed in this study, please refer to S1 Appendix under section 3.0.

To maintain a specific focus on the evaluation of artworks and their attributes while avoiding the inclusion of scale groupings found in previous studies, we excluded ratings that considered participants' elicited emotions [e.g., 59]. However, it is important to note that a clear demarcation between emotional responses and judgments is not always possible. Therefore, judgments such as "aesthetically moving" or attributes like "emotional expressiveness" were considered in our analysis. Nevertheless, we explicitly instructed participants to concentrate on the evaluation of the artwork's representations rather than their personal emotional experiences. Our study acknowledges that our selection of publications is not exhaustive. We specifically focused on studies that discussed scales related to judgments or attributes, recognizing that other research explores variables such as liking or beauty with different research objectives. In essence, our review aimed to encompass studies that delved into the intricacies of these scales rather than using ratings solely as an outcome measure for alternative research foci.

## Art judgements scale

Our review of art judgments resulted in 13 items, which we sorted along six aspects of art-judgements (see Table 1, I-VI): aesthetic, qualitative, epistemic, adverse, semantic, and preference aspects. In S1 Appendix, we provide more detail on each aspect and the items representing the 13 judgments included within each aspect.

**Table 1. Scale-set of art-judgements (targets in machine learning analysis) used in the empirical study, English version (see for German version S1 Table).**

| | Instruction | *Please provide your rating on the artwork*: |
|---|---|---|
| **Aspects** | **Scale items of art-judgments** | **Questions** |
| I. Aesthetic aspects | 1. Aesthetically moving | How aesthetically moving do you find the artwork? |
| | 2. Beauty | How beautiful do you find the artwork? |
| II. Qualitative aspects | 3. Good work of art | Do you find the painting a good artwork? |
| | 4. Creativity | How creative do you find the artwork? |
| | 5. Technical skill | How highly do you rate the artist's technical skill? |
| III. Epistemic aspects | 6. Fascinating (intellectual) | Do you find the artwork fascinating/intellectually stimulating? |
| | 7. Interesting | How interesting do you find the artwork? |
| | 8. Thought-provoking | How much does the artwork make you thoughtful? |
| IV. Adverse aspects | 9. Boring | How boring do you find the artwork? |
| | 10. Disturbing, irritating | How disturbing or irritating do you find the artwork? |
| V. Semantic aspects | 11. Familiarity | How familiar does the artwork seem to you? |
| | 12. Understanding | How much do you understand the artwork? |
| VI. Preference aspects | 13. Liking | How much do you personally like the artwork? |

## Art-attributes scale

For the selection of art attributes, we primarily followed the framework provided by 'The Assessment of Art Attributes' (AAA) by Chatterjee and colleagues [18]. Although the AAA specifically aims to advance studies in the field of neuropsychology of art, it serves as a versatile instrument for quantifying attributes of artworks and assessing differences in any context. Considering the comprehensive range of art attributes covered by the AAA, we decided to focus on this list and adapt it accordingly. The AAA includes 12 art attributes, divided into two perspectives: formal-perceptual attributes and content-representational attributes. This division we also kept in our study. The formal-perceptual attributes of the AAA consist of balance, color saturation, color temperature, depth, complexity, and brushstroke. On the other hand, the content-representational attributes encompass abstractness, animacy, emotion, realism, objective accuracy, and symbolism.

In our attribute scale set, we introduced three additional scales: (e) color variety, (g) color world, and (i) utilization of drawing area. The inclusion of these scales was motivated by the frequency with which color was addressed in the literature [13, 48–51, 60–64]. Various studies examined aspects such as color saturation, color temperature, color combination, color contrast, and specific color values. To capture the comprehensive nature of color aspects, we added (e) color variety (ranging from few to many colors) and (g) color world (ranging from dark to light colors) to our scales. Furthermore, the AAA does not include questions regarding the spatial structure or composition of artworks. To address this aspect, we introduced the scale (i) utilization of drawing area, which ranges from little utilization to full utilization of the painting area [65].

Regarding the content-representational attributes, we revised the animacy item. The decision to revise the animacy scale was based on its limited usage in other literature compared to the more frequently used attribute of (n) liveliness with poles still to dynamic.

Based on our literature review, we also introduced two additional items: (p) valence (ranging from positive to negative). This attribute has been recognized as a major dimension in various art models and as a specific factor influencing preferences for art [1, 3, 66–68]. We also

added (q) focus. The inclusion of the focus scale (ranging from much contextual information to focused content) aimed to account for potential cultural differences observed in previous research, where Asian individuals may rely more on contextual information and Western individuals focus on specific content when making judgments [69–71]. Although our participant sample is European, in outlook of future studies, we decided to include this item.

One major change was that we utilized the scale from the AAA, employing semantic differentials (opposite word pairs) for all items to represent each attribute's poles as it has been applied in past studies [5, 27, 72]. The list of art attributes can be found in Table 2.

## Methods: Rating assessment utilizing machine learning

Based on the review and established list of art judgments and attributes, we developed a study design to investigate our research question using data-driven machine learning.

### Participants

The ratings were collected from 78 psychology students at the University of Vienna (55 women, $M_{age}$ = 24.23, $SD$ = 3.45, ranging from 19 to 35, data collection took place 07.11.2019 to 13.11.2020). Each participant rated the full set of artworks (see below). All participants signed informed consent, were informed about the purpose of the study, and participated for course credit. We also asked them for their art education, their self-reported art-making experience as a hobby or professional artist, and their self-identified knowledge and interest in art [based on previously published methods by, e.g., 73]. Ethical approval was not required for the studies involving humans because ethical review and approval was not required for the study on human participants in accordance with the local legislation and institutional requirements.

**Table 2. Scales of attributes (predictors in machine learning analysis) used in the empirical study, English version (see for German version S2 Table).**

| Attributes | Scale items of attribute dimensions | Instruction | |
|---|---|---|---|
| | | **Please evaluate the artwork based on the different art attributes:** | |
| | | Negative pole (minimum) | Positive pole (maximum) |
| i. Formal-perceptual attributes | a. Visual harmony (balanced) | visual harmony, proportional | peculiar, strange shapes |
| | b. Depth | two-dimensional | three-dimensional |
| | c. Complexity | simple | Complex |
| | d. Color saturation | soft, pastel | intense, strong |
| | e. Color variety | few colors | many colors |
| | f. Color temperature | warm colors | cold colors |
| | g. Color world | dark color world | light color world |
| | h. Brushwork | fine brushwork | rough brushwork |
| | i. Utilization of drawing area | little utilization of the painting area | full utilization of the painting area |
| ii. Content-representational attributes | j. Abstractness | representative | Abstract |
| | k. Imaginativeness | realistic content/topic | imaginary, unreal, fantastic |
| | l. Symbolism (ambiguity) | distinct/clear (unambiguous interpretation) | symbolic (more space for interpretation, ambiguous) |
| | m. Accurate object representation | photorealistic | painterly |
| | n. Liveliness/animation | dynamic | Still |
| | o. Emotional expressiveness | emotionless | emotionally loaded |
| | p. Valence | negative valence | positive valence |
| | q. Focus | much context/environment in the image | focused content |

All participants provided their written informed consent to participate in this study. The study corresponds with the ethical standards of the Declaration of Helsinki and the ethical regulations at the University of Vienna. No retrospective data was included in the study.

The number of participants followed best practice in artwork rating studies [36, sample of at least 10 judges provide stable scoring with good reliability, e.g., 74–76] and best practice for high feature predictability using machine learning. Because precise sample size justifications for complex, high-dimensional, multivariable models from the machine learning field have not yet been standardized, our power analysis follows the most recent suggestions [77, 78]: (1) A minimum of 50 samples are required to start any meaningful machine learning based data analysis [79]. (2) 10 to 20 samples per degree of freedom (predictor) is reasonable, which would lead to a total number of 170 to 340 ratings (samples) required in the present case [80, 81]. (3) A power analysis using the two-tailed student's $t$-test, with an alpha of 0.05 and a power of 0.8 suggests that 620 ratings (samples) would be required to detect small effects of Cohen's d 0.2 [82]. Generally, as in other statistical data analysis methods, more samples allow finding smaller effects. Based on these considerations and available resources, we collected a total number of 4206 ratings (samples, 78 participants evaluating 54 images, 6 picture ratings were missing due to recording issues).

## Materials

We selected 54 images of Western paintings from the Vienna Art Picture System [VAPS, 83] as stimuli for our study (see for full list of artworks S3 Table). The VAPS provided a suitable platform for stimulus selection, offering a diverse range of 999 fine art paintings with various motives, styles, and time epochs, along with pre-existing rating scores (liking, valence, arousal, complexity, familiarity) for each stimulus.

To ensure variation in the artworks, we included three subsets of depicted motifs (portrait, landscape, still life) and three style categories (representative, impressionistic, abstract art). The number of images was balanced across the depicted motifs and styles. We conducted a Wilk-Shapiro test (see S4 Table) to assess the normal distribution of the provided ratings for the selected stimuli. Overall, we observed normal distributions in most categories, except for familiarity ratings, which showed significant differences across all categories due to the inclusion of famous and lesser-known artworks and considering pre-ratings were made by art novices in the VAPS. Additionally, more representative art was rated significantly more positively in valence and arousal, possibly due to the higher familiarity with realistic artworks.

Despite these limitations, we deemed them acceptable given the normal distributions observed in other categories and the unavailability of other stimulus databases with pre-existing ratings.

## Procedure

The testing took place in a laboratory room equipped with four workstations for simultaneous participant testing. Participants provided written informed consent and were instructed to rate visual artworks using a series of scales. They were given the freedom to take as much time as needed, with an average completion time of 90 to 120 minutes. The artworks were displayed on a 19" monitor (Iiyama ProLite B1906S, 1280x1024, 60Hz), with a maximum dimension of 500 pixels and participants seated approximately 50 cm away from the screen. The artworks and ratings were presented using the program SoSci-Survey on a University of Vienna server [84]. Each artwork was shown individually centered on the screen with a white background, while the scales were displayed below. The order of artwork presentation was randomized

between participants. The scales and items were randomized between each trial (within participant) to avoid rating sequence effects.

Two types of scales were used: (1) items of art judgments rated on a 100-point Likert-type scale from "not at all" to "very much" using a slider, and (2) items of art attributes presented as opposite pairs (semantic differentials) on a 100-point Likert scale. The slider had a middle position at the 50-point mark. The full list of scales in English can be found in Tables 1 and 2 (for German versions see S1 and S2 Tables).

## Data analysis: Multivariable regression

The multivariable statistical regression analyses were performed using Python v3.10.5 and the machine-learning framework scikit-learn library v1.1.1 [79]. Each analysis consisted of three parts: (1) prediction model training, (2) generalizability testing, and (3) model analysis.

1. Gradient Boosted Decision Tree (GBTD) models were used for the regression task because they are computationally efficient and highly accurate [17]. In addition, GBTD models can model non-linear associations and interactions between predictors and target variables. They are also robust to multicollinearity and outliers in the data. The predictors used in the models included a total of 17 attributes. The 13 art judgments were the prediction targets. Our model selection was primarily driven by the findings of recent research, such as the work by Grinsztajn and colleagues [85], which suggests that tree-based models often outperform deep learning on typical tabular data like ours.

2. To assess the performance of the models on unknown data (i.e., how well the predictions generalize), a nested cross-validation (CV) procedure was employed [86]. CV implements repeated splits of the data into training and testing set. A group-controlled shuffle-split scheme was used in the main (outer) CV loop to control for participant-related clusters (20% of participants in the testing set, 80% of participants in the training set, 50 repetitions). In each repetition, the training set was used for data scaling (standardization) and model complexity tuning. A nested (inner) CV procedure (20% of participants in the testing set, 80% of participants in the training set, 14 repetitions) was used to tune the complexity parameters of the model. To find the best performing parameters, a sequential Bayesian optimization procedure in combination with a shuffle-split scheme was used (400 repetitions, 200 initial points; BayesSearchCV, scikit-optimize, v0.9, learning_rate 1e-3 to 1e0, n_estimators 1 to 3000, num_leaves 2 to 1000, min_child_samples 1 to 1000, colsample_bytree 1e-3 to 1e0, reg_alpha 1e-3 to 1e3, extra_trees True/False). Thse parameters were used in to train regressor models in the main CV loop, all other parameters were left to default. The models were subsequently tested on the respective testing set of the main CV loop. The testing set was explicitly not used in the inner CV loop. Regression performance was evaluated using two metrics: (i) the prediction coefficient of determination ($R^2$), and (ii) the mean absolute error (MAE) [87]. $R^2$ can have values between minus infinity and 1, with a value of 0 indicating a performance as good as using the average target value as a predictor (the trivial predictor), and a value of 1 indicating no error at all. It is worth noting that the prediction $R^2$ will be smaller than $R^2$ values of conventional statistical models because the prediction $R^2$ measures prediction performance for unknown data, rather than post-hoc model fit [87]. The MAE reflects the average error made at each prediction and is not normalized; thus, the error is measured with the scale of the dependent variable, the prediction targets.

3. To assess the importance of individual predictors for the model's performance, we used SHAP (SHapley Additive exPlainations) [88, 89]. SHAP is a method from interpretable

machine learning that is based on Shapley values, a method from cooperative game theory. It measures the contributions of each predictor to the model's performance. Pooling those contributions over many predictions allows for a comprehensive analysis of the importance of individual predictors for the regression task [88, 89].

## Data analysis: Statistical tests

Statistical significance of the prediction $R^2$ and MAE metrics as well as of the predictor's importance's was assessed using a modified $t$-test that takes the sample dependence due to CV into account [90, 91]. $T$-test results with Bonferroni corrections are presented together with the ones without correction.

We note that detailed results on how each predictor influences the prediction results shown below are available from this link: https://github.com/univiemops/art-rating-prediction.

## Results

We applied machine learning to predict 13 art judgments based on ratings of visual art attributes, in order to identify which attributes (if any) contribute to the predictions. We computed the multivariate models and evaluated their performance. The GBDT models provided a good fit within the range of conventional cutoff values (as can be seen in Table 3). In total, all 13 judgments could be significantly predicted by the attributes. The results showed average absolute errors of 14.85 (SD = 1.20) for the lowest *disturbing/irritating* and 21.48 (SD = 0.83) for the highest, *familiarity*, within a range of 1 to 101. The prediction coefficient of determination $R^2$ ranged between 0.02 for *familiarity*, 0.14 for *good work of art* as the lowest $R^2$ value, and 0.31 for *creativity* and 0.32 for *disturbing/irritating* as the highest $R^2$ value. This indicates that on average, the model explains 31% of the variance in the *creativity* ratings and 32% of the variance in *disturbing/irritating* ratings.

**Table 3. GBDT regressor prediction results of all art judgment ratings.** 50 repetitions from cross-validation.

| Scales of art-judgments | Mean Absolute Error (MAE) ± standard deviation | p value of obtained MAE bigger or equal to MAE from shuffled labels data | Coefficient of determination ($R^2$) ± standard deviation | p value of obtained $R^2$ smaller or equal to $R^2$ from shuffled labels data |
|---|---|---|---|---|
| Aesthetically moving | 19.79 ± 1.33 | *p* = 0.009 | 0.15 ± 0.04 | *p* < 0.001 |
| Beauty | 20.09 ± 1.10 | *p* < 0.001 | 0.20 ± 0.06 | *p* < 0.001 |
| Good work of art | 19.02 ± 1.21 | *p* < 0.001 | 0.14 ± 0.06 | *p* < 0.001 |
| Creativity | 17.34 ± 1.20 | *p* < 0.001 | 0.31 ± 0.04 | *p* < 0.001 |
| Technical skill | 16.66 ± 1.48 | *p* < 0.001 | 0.16 ± 0.09 | *p* < 0.001 |
| Fascinating (intellectual) | 18.62 ± 1.24 | *p* < 0.001 | 0.22 ± 0.04 | *p* < 0.001 |
| Interesting | 19.55 ± 1.38 | *p* < 0.001 | 0.25 ± 0.05 | *p* < 0.001 |
| Thought provoking | 18.18 ± 1.14 | *p* < 0.001 | 0.22 ± 0.05 | *p* < 0.001 |
| Boring | 20.77 ± 1.30 | *p* < 0.001 | 0.22 ± 0.05 | *p* < 0.001 |
| Disturbing, irritating | 14.85 ± 0.83 | *p* < 0.001 | 0.32 ± 0.06 | *p* < 0.001 |
| Familiarity | 21.48 ± 1.18 | *p* = 0.002 | 0.02 ± 0.04 | *p* = 0.001 |
| Understanding | 19.23 ± 1.40 | *p* < 0.001 | 0.24 ± 0.08 | *p* < 0.001 |
| Liking | 21.15 ± 1.26 | *p* < 0.001 | 0.18 ± 0.05 | *p* < 0.001 |

The results of our machine learning predictions for aesthetic aspects showed prediction-based coefficients $R^2$ in the range of 0.15 for *aesthetically moving* to 0.20 for beauty. Qualitative aspects vary strongly, with 14% explained variance for *good work of art*, *creativity* showing the highest explained variance at 31% and with 16% variance for *technical skill*. Similarly, general aspects showed low explanation strength for *familiarity* ($R^2 = 0.02$) but relatively high explanation strength for *understanding* (24% variance). Epistemic aspects, such as *fascinating*, *interesting*, and *thought-provoking*, provide a relatively similar explanation of variance (>22%). Adverse aspects also show a higher variance explanation, with 22% for boring and 32% for *disturbing/irritating*. Last, *liking* had an average variance explanation of 18%.

Furthermore, we analyzed the contributions of individual factors to the model's performance, i.e., which factors are important to predict the art-judgments ratings. In Figs 1 and 2 we show two examples of judgments, aesthetically moving and liking, representing two examples of lower degree of attribute explanation (lower $R^2$); Figs 3 and 4 show the results of the judgments creativity and disturbing/irritating as examples for high explained variance attributes.

Our analysis revealed several key findings regarding the importance of different attributes in art judgments (see Figs 1 to 2 as well as https://github.com/univiemops/art-rating-prediction for full list of plots). See for overview of attribute's average rank Fig 5.

Firstly, emotional expressiveness and valence emerged as the most significant attributes across multiple judgments. Emotional expressiveness exhibited high importance for judgments related to aesthetically moving, beauty, good work of art, creativity, fascinating, interesting, thought-provoking, boring, and liking, with mean absolute values ranging from 3.37 to 7.25. Similarly, valence played a crucial role in judgments of aesthetically moving, beauty, good work of art, disturbing/irritating, familiarity, and liking, with mean values ranging from 3.35 to 6.24. Additionally, symbolism was a prominent attribute for creativity, epistemic aspects (fascinating, interesting, thought-provoking), adverse aspects (boring, disturbing/irritating), and the semantic aspect understanding, with mean values ranging from 3.81 to 7.81. Furthermore, within the content-representational attribute group, abstractness was found to be particularly important for the judgment of creativity and, to a lower extend, disturbing/irritating. Imaginativeness contributed to creativity and understanding, and to a lesser degree to adverse and epistemic aspects.

In the formal-perceptual features group, visual harmony (balance) demonstrated relevance for multiple judgments. It made notable contributions to aesthetically moving (moderate mean SHAP value: 2.77) and beauty (mean SHAP value: 5.26), as well as good work of art (mean SHAP value: 3.32) and liking (mean SHAP value: 4.14) among others.

Depth contributes to every judgment, however considering rank was often on the fifth or sixth ranks. The attribute was found to be significant for aesthetically moving, although with a relatively moderate mean SHAP value of 2.63. It also showed some relevance to technical skill, albeit with a modest mean SHAP value of 2.84. Interestingly, brushstroke was mainly contributing to technical skill (mean SHAP value: 3.77).

Complexity exhibited relevance for qualitative aspects, specifically good work of art, creativity, and technical skill. Further, complexity contributed with the lowest mean SHAP value of 2.79 for liking and the highest 4.68 for boring. Considering color attribute in general, hardly any showed strong impact on the judgments. In sum the results indicate that content-representational attributes were the dominant contributors often gaining ranks one to three with higher mean SHAP value compared to those in the formal-perceptual group.

Furthermore, we examined the specific attributes for judgments depicted in Figs 1 to 4. For the judgments of aesthetically moving and liking, the order of importance for attributes was largely similar, with emotional expressiveness, valence, and visual harmony consistently

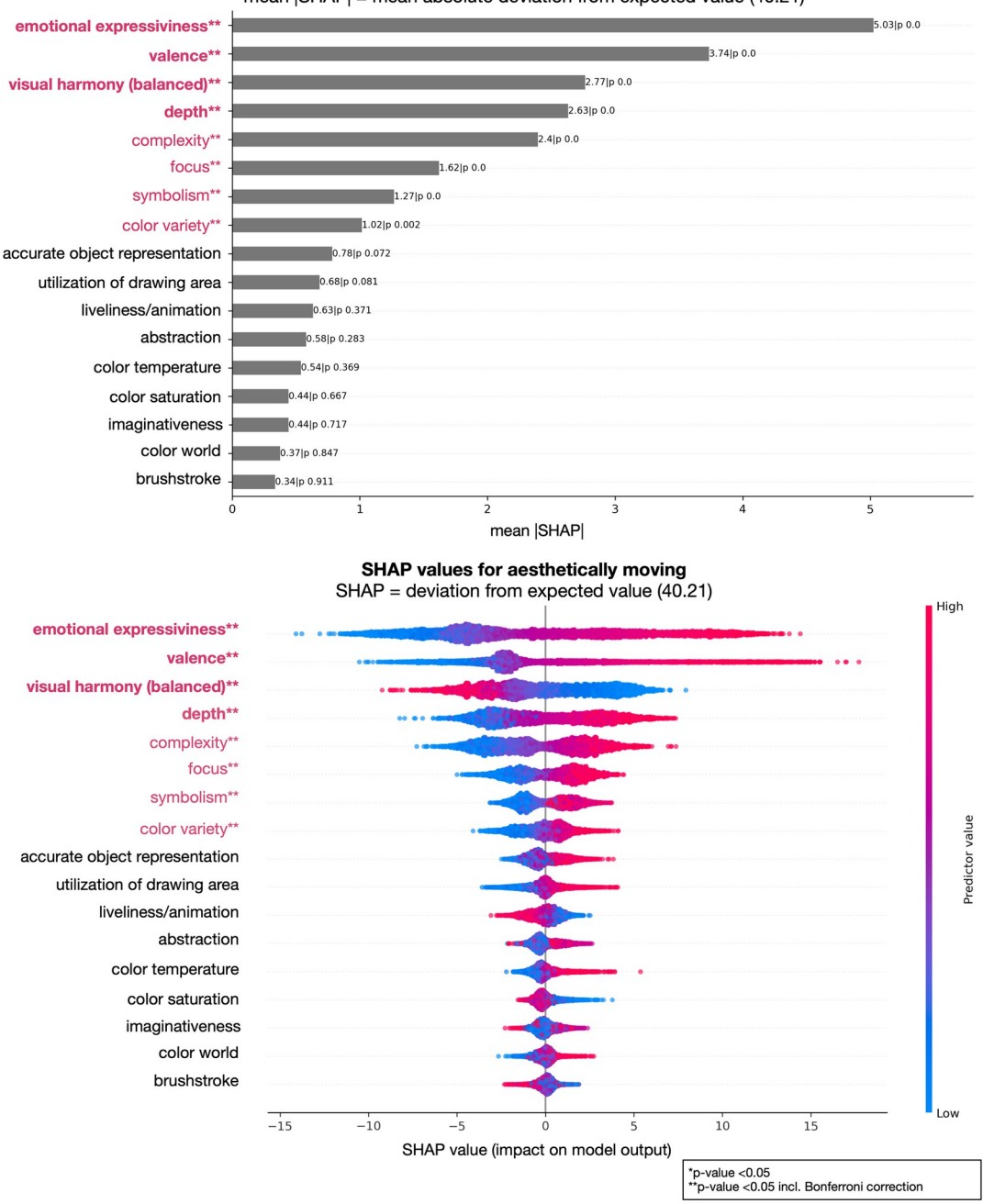

**Fig 1.** Art-attributes importance's for predicting aesthetically moving: Top plot shows mean absolute SHAP (SHapley Additive exPlanations) values for all 17 attributes in the model. Bottom plot displays SHAP values directly (color code: low = blue, red = high, and showing non-linearity of associations) and offers a detailed view of the individual impacts of predictors on specific predictions. **Bold and red attributes represent the most important attributes considering mean SHAP values; **red attributes significant with Bonferroni correction, however with a lesser influence considering mean SHAP values; *significant attributes before Bonferroni correction.

identified as the top three attributes. However, some attributes that remained significant after Bonferroni correction, but had relatively low mean SHAP values ($< 2.70$, see for detailed results https://github.com/univiemops/art-rating-prediction), did not play a substantial role in

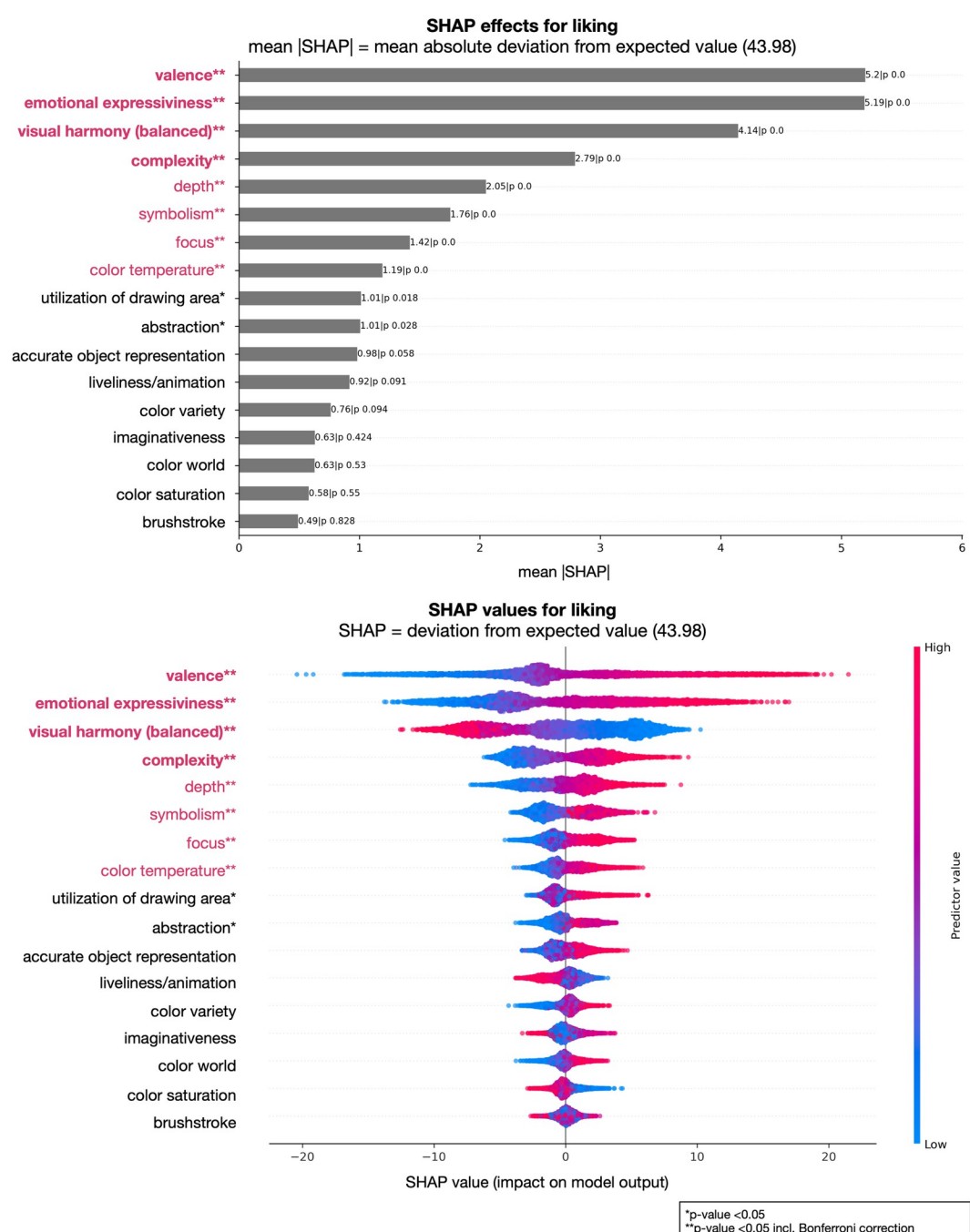

**Fig 2.** Art-attributes importance's for predicting liking: Top plot shows mean absolute SHAP (SHapley Additive exPlanations) values for all 17 attributes in the model. Bottom plot displays SHAP values directly (color code: low = blue, red = high, and showing non-linearity of associations) and offers a detailed view of the individual impacts of predictors on specific predictions. **Bold and red attributes represent the most important attributes considering mean SHAP values; **red attributes significant with Bonferroni correction, however with a lesser influence considering mean SHAP values; *significant attributes before Bonferroni correction.

these judgments. Herein, specifically, depth and complexity exhibited opposite orders of importance in the two judgments.

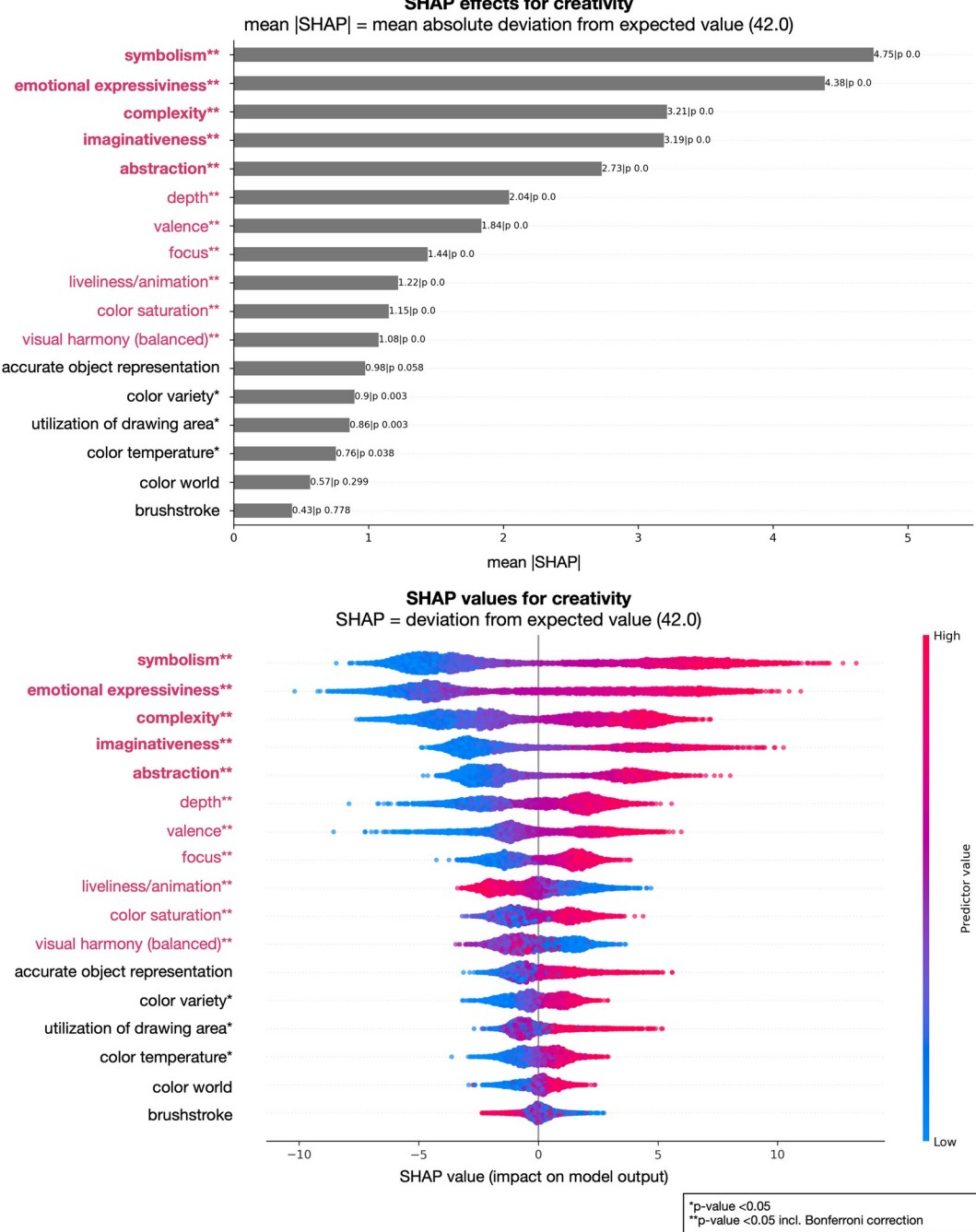

**Fig 3.** Art-attributes importance's for predicting creativity: Top plot shows mean absolute SHAP (SHapley Additive exPlanations) values for all 17 attributes in the model. Bottom plot displays SHAP values directly (color code: low = blue, red = high, and showing non-linearity of associations) and offers a detailed view of the individual impacts of predictors on specific predictions. **Bold and red attributes represent the most important attributes considering mean SHAP values; **red attributes significant with Bonferroni correction, however with a lesser influence considering mean SHAP values; *significant attributes before Bonferroni correction.

When considering the attributes for aesthetically moving and liking with Bonferroni correction, additional attributes emerged as important factors. For aesthetically moving,

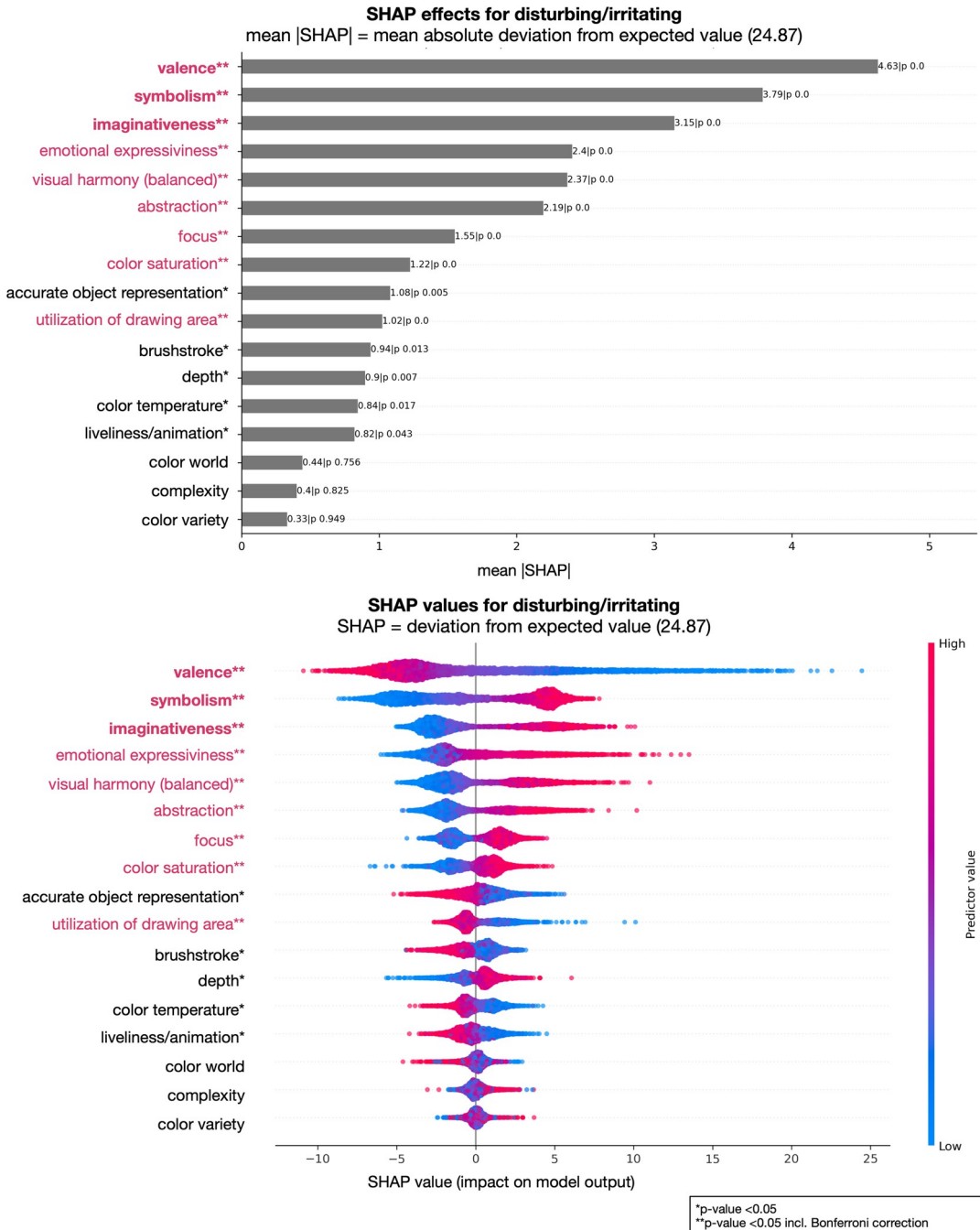

**Fig 4.** Art-attributes importance's for predicting disturbing/irritating: Top plot shows mean absolute SHAP (SHapley Additive exPlanations) values for all 17 attributes in the model. Bottom plot displays SHAP values directly (color code: low = blue, red = high, and showing non-linearity of associations) and offers a detailed view of the individual impacts of predictors on specific predictions. **Bold and red attributes represent the most important attributes considering mean SHAP values; **red attributes significant with Bonferroni correction, however with a lesser influence considering mean SHAP values; *significant attributes before Bonferroni correction.

symbolism and color variety were notable attributes; while symbolism, focus, color temperature, utilization of drawing area, and abstractness were notable further attributes for liking.

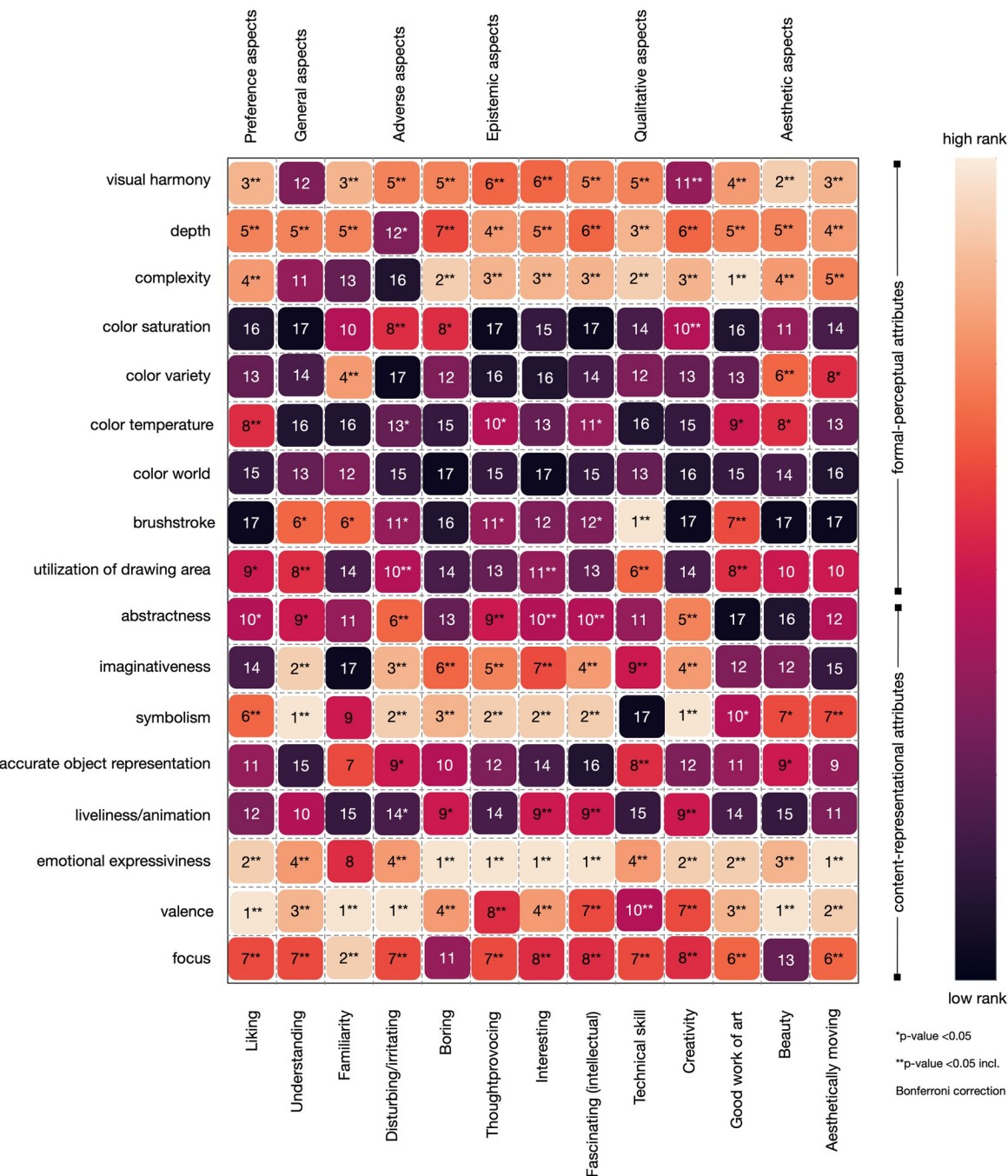

**Fig 5. Rank based on mean SHAP-values for each attribute in association to the art judgments; \*_p_-value <0.05; \*\*_p_-value <0.05 incl.** Bonferroni correction.

The judgment of creativity presented a different pattern, with symbolism identified as the most important attribute, followed by emotional expressiveness, complexity, imaginativeness, and abstractness. Depth, valence, focus, liveliness/animation, color saturation, and visual harmony demonstrated lesser importance based on mean SHAP values (see for detailed results https://github.com/univiemops/art-rating-prediction). Without Bonferroni correction, utilization of drawing area and color variety were found to contribute to creativity.

In the case of the judgment disturbing/irritating, valence emerged as the most important attribute, followed by symbolism and imaginativeness. Emotional expressiveness, visual harmony, abstractness, focus, color saturation, and utilization of drawing area showed lesser importance based on mean SHAP values (see Fig 4). Without Bonferroni correction, accurate object representation, depth, color temperature, and animation/liveliness were found to be contributing factors.

Additionally, in Figs 1 to 4 show on the bottom a visual representation of the impact of attributes on each judgment. This representation not only indicates the directionality of the impact (positive or negative) but also demonstrates the degree of influence, with attributes having varying levels of impact from high to low (color coding red to blue, respectively). Moreover, the non-linear associations between attributes and judgments are evident for single cases, further highlighting the complex and nuanced nature of the relationships; a relationship between predictors and target, which might have been overseen when using linear models.

In addition to the primary analyses, we conducted further model comparisons and an examination of predictor importance, the details of which are presented in S2 Appendix. Notably, we compared the performance of Gradient Boosting Decision Trees (GBDT) with various Generalized Linear Models (GLMs), including models without regularization (GLM-OLS), with regularization (GLM-Elastic-Net), and enhanced with polynomial features (GLM-Elastic-Net-poly-feat). Our findings highlighted GBDT's superior predictive accuracy, underscored the value of incorporating non-linear terms and regularization in modeling, and confirmed GBDT's leading position in terms of average $R^2$ values across 13 judgments. Additionally, we analyzed the most significant predictors identified by both GBDT and GLM-Elastic-Net linear regression, revealing a remarkable consensus on the top predictors despite the intrinsic differences between linear and non-linear modeling approaches. For those interested in a deeper exploration of all predictors and their comprehensive rankings across different prediction targets and methods, we have made the data available on our GitHub repository (https://github.com/univiemops/art-rating-prediction).

Last, we provide a correlation heatmap (see S2 Fig). These heatmap reveal the following pattern: although there are correlations among the art attributes themselves and the art judgments themselves, there are minimal correlations between the art judgments and the attributes. In other words, while the attributes may exhibit relationships with each other, as do the judgments, the attributes do not strongly correlate with any judgments. These finding support our machine learning approach and point to the complex and multifaceted nature of art evaluation.

## Discussion

The present study aimed to investigate the relationship between art judgments and art attributes using a machine learning framework. Through non-linear modeling techniques, we aimed to overcome the limitations of traditional linear models and gain a deeper understanding of the complex dynamics underlying art evaluation [16, 17, 38, 40–43]. Our findings are based on prior research focusing on creativity judgments only [12]. We now broadened and updated the analysis to in total 13 art judgements using 17 attributes as predictors.

Our study revealed that specific art attributes significantly predicted art judgments, with content-representational attributes playing a more prominent role than formal-perceptual attributes addressing individual judgments. This finding aligns with previous research suggesting that the subjective interpretation and evaluation of artworks rely heavily on the content and meaning conveyed by the artwork [3, 5, 12, 14, 92]. Emotional expressiveness, valence, and symbolism emerged as particularly important attributes, appearing in the

predictions for multiple judgments. These attributes capture the strength of the artist to express emotion, the positivity/negativity, and the level of ambiguity/symbolistic meaning associated with artworks [1, 35, 66, 83].

The non-linear relationships observed between art attributes and judgments further highlight the complexity of art evaluation. Some cases exhibited sudden inclines or declines around the midpoint of the scales, suggesting that the impact of an attribute on a judgment may vary depending on its magnitude. This finding underscores the need to consider the multidimensional nature of art evaluation and an interplay between different attributes [37, 38, 42, 43, 93]. This provides a fascinating implication for future studies suggesting that judgments are not solely determined by the presence or absence of specific attributes but rather by the dynamic interaction between various attributes.

Our findings considering the non-linear associations with sudden in- or declines hint to an intriguing pattern, which might best be understood as an aesthetic threshold [see also discussion in, 12]. This phenomenon might hint to a critical point at which the perception and evaluation of an artwork shift significantly, a concept also suggested in psychophysics where perceptual changes do regularly scale non-linear with stimulus variation. Such aesthetic thresholds have been discussed to occur around the midpoint of the rating scales for several art attributes, where Berlyne's studies and later work suggest with some attributes, such as complexity, a non-linear relationship [5, 27, 32, 37, 94, 95].

Importantly, our study contributes to addressing the gap in the literature regarding the relationship between art judgments and attributes. First, previous research often focused on a limited number of judgments and attributes without systematically examining their connections. Second, one key distinction that past literature has often overlooked is the differentiation between art judgments and art attributes. Art judgments refer to the evaluative opinions and assessments individuals make about artworks, encompassing concepts such as beauty, liking, interestingness, and thought-provoking. On the other hand, art attributes are specific qualities or features of artworks that individuals use to substantiate their evaluative arguments. These attributes provide a foundation for articulating and validating subjective evaluations, such as colorfulness, emotional expressiveness, imaginativeness, and complexity. The correlation heatmap (see S2 Fig) support this notion, showing very low correlations between judgment and attributes. By failing to dissociate between judgments and attributes, previous research has not fully explored the underlying factors and mechanisms that shape art evaluations.

Third, future research should also consider the difference between evaluating an artworks representation and the focus of elicited emotion within the person. Although, naturally, emotion and evaluation cannot be departed, the focus of judgments to be provided can be framed, which we tried to respect in our study. The task given should be clearly stated in the participant instructions. However, we acknowledge that also we cannot prove if we were fully successful within our study; future studies could investigate differences in task instructions (focus on artwork evaluation or on own emotional experience).

Considering the attributes importance we found a dominance of higher importance of content-representational attributes in predicting art judgments. This suggests the importance of meaning and emotional resonance in aesthetic experiences. Artworks are a medium that can represent and mirror emotions, convey symbolic messages, and resonate with personal experiences. This is a fundamental aspect of art appreciation supporting also new frameworks in the field considering this interplay of knowledge, emotion, and the dynamic interaction predicting the value of an experience [67, 96, 97]. Therefore, our research approach should also be extended to investigate this interplay, starting with the association between the strength of emotional expressiveness and the elicitation of emotions and connected to meaning making

based on experience [98]. Our results also support further study of aesthetic experiences in visual art research specifically, such as awe and wonder [99–102].

Our findings show that art judgments are not solely based on formal aspects such as color or composition but are strongly influenced by the content and meaning portrayed by the artworks, however, it is worth noting that some formal-perceptual attributes also made significant contributions to the predictions, albeit to a lesser extent than content-representational attributes.

Considering formal-perceptual features, especially visual harmony, depth, but also complexity were significant contributors (see for example, Figs 1 and 2). Complexity emerged as the most relevant formal-perceptual attribute, particularly in judgments related to boring (simple images were more boring), interestingness (low complex artworks were less interesting), and all judgments from the qualitative aspects (good work of art, creativity, and technical skill) showing that high complex art images received higher ratings. Note, for good work of art complexity was the most important contributor. These finding is consistent with previous research highlighting the role of complexity in art evaluation and the influence of cognitive processes on aesthetic experiences [37, 94, 103]. Visual harmony also appeared as a predictor for the judgments aesthetically moving, beauty, and liking indicating its relevance to the very specific evaluation of appreciating aesthetic elements in art. Here, we should highlight our cohort, where laymen might orient their liking more on aesthetic appeal; while art experts, for example, might use other attributes for their personal preferences.

The low contributions from certain attributes, such as color saturation, color variety, and other color aspects, suggests that these attributes may have limited impact on art judgments in our art novice study cohort. These finding challenges previous assumptions about the universality and centrality of these attributes in art evaluation [65, 104], although still may have relevance in cross-cultural comparison studies [69, 105] or comparisons with art experts [106–108]. The absence of some attributes also highlights the need for a more nuanced understanding of the contextual factors that influence the importance and relevance of different attributes in art appreciation.

In sum, our research reveals a complex interplay of attributes contributing to art judgments, were sometimes the same attributes have high impact for different judgments. Apparently, it lies in the nuances and interplay of the other attributes, that determine the judgments differently. Another assumption would be to assume that art novices might treat some judgments as being the same (for example, beauty and liking). To support this interpretation more research is necessary in diverse cohorts and certainly also employing more within subject comparison study designs.

Our study also expands on previous research by incorporating machine learning methodologies and non-linear modeling techniques [12, 48–52, 109]. By employing machine learning techniques, we were able to identify a comprehensive set of attributes that significantly contribute to the predictions of art judgments [16, 17, 40–43]. This data-driven approach provides a more nuanced understanding of the factors that shape art evaluations and highlights the interplay between different attributes.

This approach allowed us to capture complex relationships and patterns that may have been overlooked by traditional linear models [38, 40]. By leveraging machine learning algorithms, we were able to construct data-driven models that provide a more comprehensive understanding of art evaluation. This approach has the potential to uncover hidden patterns, regularities, and potential inconsistencies within art judgments, contributing to the advancement of knowledge in aesthetics.

The findings of our study have implications for various domains, including art education, curatorial practices, and the development of computational models of aesthetics. By gaining a

deeper understanding of the factors that shape art judgments, educators, curators, and online platforms can design more engaging and meaningful art experiences, adapted to the target group. They can emphasize the importance of emotional expressiveness, symbolism, and other content-representational attributes in fostering aesthetic engagement and appreciation, addressing for laymen rather the whole artwork than individual elements [4]. Furthermore, our findings provide valuable insights for the development of computational models that simulate human-like art evaluations. By incorporating the identified attributes and their non-linear relationships, these models can generate more realistic and nuanced judgments of artworks.

Despite the valuable insights provided by our study, there are also limitations. First, we only tested art novices, which limits the generalizability, especially to expert art evaluators or individuals with different cultural backgrounds. Future research could explore the potential differences in the relationship between art judgments and attributes across diverse populations, employing the same procedures. Second, the selection of artworks in our study was limited to artworks of Western culture, which restricts the generalizability of the results to other artistic traditions and cultures. Future studies could incorporate artworks from a wider range of cultural contexts to examine the cross-cultural variations in art judgments and their associated attributes. Lastly, the use of machine learning techniques introduces the challenge of interpretability. While these models provide accurate predictions, understanding the underlying mechanisms and causal relationships between judgments and attributes may require additional analysis and experimentation.

In conclusion, our study provides valuable insights into the complex interplay between art judgments and attributes. By employing machine learning techniques and non-linear modeling, we uncovered the significance of content-representational mainly and a view formal-perceptual attributes in shaping art judgments. Among these attributes, complexity and visual harmony are certainly the two most interesting ones. Our findings highlight the non-linear nature of the relationships between attributes and judgments and emphasize the importance of emotional expressiveness, valence, symbolism, and complexity in art appreciation. These insights contribute to a deeper understanding of the multidimensional nature of aesthetic experiences and have implications for art education, art practices, and the development of studying non-linear human behavior in art research. In addition, we also want to highlight the apparent relevance of level of symbolism (interpretability), meaning, and emotional expressivity. Future research can build upon these findings by exploring the cross-cultural variations in art judgments, focus on one judgement or attributes solely, investigate the role of individual differences, and further refining the understanding of the underlying mechanisms of art evaluation.

## Supporting information

**S1 Table. Scales of art-judgements (targets in machine learning analysis) used in the study, German version.**
(PDF)

**S2 Table. Scales of art attributes (predictors in machine learning analysis) used in the study, German version.**
(PDF)

**S3 Table. Full list of stimulus-set rated including name of artist, title, style, and depicted motif.** ID represents the VAPS-identification code.
(PDF)

**S4 Table. Stimulus selection of VAPS using Wilk-Shapiro test for ensuring best possible normal distribution of VAPS ratings while including images.**
(PDF)

**S1 Fig. Procedures of systematic literature review.**
(TIF)

**S2 Fig. Correlation heatmap of art judgments and attributes.**
(TIF)

**S1 Appendix. Review of art related judgments and attributes, including list of judgements descriptions** [1, 3, 5, 12, 13, 15, 18, 23, 24, 27–32, 34–37, 46–50, 59–65, 67–69, 72, 73, 94, 95, 98–102, 105, 108, 110–179].
(DOCX)

**S2 Appendix. Model comparisons and an examination of predictor importance.**
(DOCX)

## Author Contributions

**Conceptualization:** Blanca T. M. Spee, Helmut Leder, Jan Mikuni, Matthew Pelowski.

**Data curation:** Blanca T. M. Spee, Jan Mikuni.

**Formal analysis:** Blanca T. M. Spee, Jan Mikuni, Frank Scharnowski, David Steyrl.

**Investigation:** Blanca T. M. Spee.

**Methodology:** Blanca T. M. Spee.

**Project administration:** Blanca T. M. Spee.

**Software:** David Steyrl.

**Supervision:** Helmut Leder, Frank Scharnowski.

**Validation:** Blanca T. M. Spee, David Steyrl.

**Visualization:** Blanca T. M. Spee.

**Writing – original draft:** Blanca T. M. Spee, Helmut Leder, David Steyrl.

**Writing – review & editing:** Blanca T. M. Spee, Helmut Leder, Jan Mikuni, Frank Scharnowski, Matthew Pelowski, David Steyrl.

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
