## [Decision Letter · Decision Letter 0]

20 Nov 2023

PONE-D-23-31191Using Machine Learning to Predict Judgments on Western Visual Art Along Content-Representational and Formal-Perceptual AttributesPLOS ONE

Dear Dr. Spee,

Thank you for submitting your manuscript to PLOS ONE. After careful consideration, we feel that it has merit but does not fully meet PLOS ONE’s publication criteria as it currently stands. Therefore, we invite you to submit a revised version of the manuscript that addresses the points raised during the review process.

We look forward to receiving your revised manuscript.

Kind regards,

Maja Vukadinovic

Academic Editor

PLOS ONE

6. Please include the reference section of your manuscript. 

7. Please remove your figures from within your manuscript file, leaving only the individual TIFF/EPS image files, uploaded separately. These will be automatically included in the reviewers’ PDF.

8. Please upload a copy of Figure 4, to which you refer in your text on page 30. If the figure is no longer to be included as part of the submission please remove all reference to it within the text.

10. Please upload a copy of Figure S2, to which you refer in your text. If the figure is no longer to be included as part of the submission please remove all reference to it within the text.

Reviewer's Responses to Questions

**Comments to the Author**

1. Is the manuscript technically sound, and do the data support the conclusions?

Reviewer #1: Yes

Reviewer #2: Yes

2. Has the statistical analysis been performed appropriately and rigorously? 

Reviewer #1: Yes

Reviewer #2: Yes

3. Have the authors made all data underlying the findings in their manuscript fully available?

Reviewer #1: Yes

Reviewer #2: Yes

4. Is the manuscript presented in an intelligible fashion and written in standard English?

Reviewer #1: Yes

Reviewer #2: Yes

5. Review Comments to the Author

Reviewer #1: The authors of this paper have conducted a comprehensive review to identify the key attributes of arts and art judgments categories. They employed a nonlinear machine learning technique to demonstrate that behaviorally collected art judgments can be predicted using these key attributes, which were also gathered behaviorally. Additionally, the authors analyzed the contributions of each attribute to the prediction of various categories of art judgments.

The study exhibits a meticulous approach to the literature review and effectively applies machine learning methodology. However, some crucial questions remain to be addressed. It is imperative to conduct a model comparison with other models to substantiate the authors’ findings and establish the necessity of nonlinearity in their approach. Specifically, I propose a comparative analysis that includes a GLM model incorporating attributes, their interactions and their quadratic terms to predict judgments, in order to gauge the superiority of the GBDT model. Additionally, it would be valuable to assess the significance of each attribute in predicting judgements by comparing with other feature selection methodologies, such as ridge regression. Such comparisons are necessary to validate the claims made in this study regarding the importance of individual attributes. For example, the disparities in finding between this paper and Iigaya’s work, where the significance of concreteness (equivalent to abstraction in this work) and valence yield opposing results (figure 1 and 3). These discrepancies, albeit stemming from varying features, stimuli, and methodologies, necessitate comparisons with other models for robust validation.

While the paper demonstrated the importance of individual attributes, it falls short of elucidating how the manipulation of these attributes impact art judgments in an interpretable manner. I recommend conducting a comprehensive analysis of the GBDT model’s predictions while varying input attributes, as this would provide insight into the relationship between the art attributes and the art judgments. (e.g., how the liking rating changes when the abstraction level goes up)

Furthermore, it would be beneficial to include calculations for the noise ceiling to assess the theoretical upper limit for predicting art judgement based on the attributes. This evaluation will offer a deeper understanding of the model’s capabilities and the limit of the machine learning based approach in this literature.

In conclusion, although this research is meticulously designed and executed, addressing the aforementioned points will enhance the study’s robustness.

Reviewer #2: I have reviewed this paper and find it highly interesting and well done. I have however a number of recommendations that would make the paper better. Even though none of my concerns were large enough to warrant the answer "No" to questions 1-4, they are together larger than warranting just cosmetic changes, and thus my recommendation becomes "Major revision".

1: The topic of the paper, to investigate the correlations between subjective attributes of an artwork and value-based judgements of the same artwork, is highly interesting and relevant. A data-driven method based on recordings of user studies, and a regression-based Machine Learning (ML) method has been used. The method is clearly described and reproducable, and the results are rigorously analyzed and discussed. All conclusions in the (extremely ambitious and rigorous) Discussion section are well underbuilt by the experimental results.

Recommendation:

With a background in ML, I have some suggestions. I firstly would like to emphasize that I do not ask for new or changed experiments, as the currently applied ML method is adequate for your study.

However, I have requests concerning the framing and references to the state of the art. I recommend a newer ML introduction reference than Breiman, e.g. Alpaydin's book from 2020: https://ds.amu.edu.et/xmlui/bitstream/handle/123456789/6943/Introduction%20to%20Machine%20Learning.pdf. In general, this field is moving forward so fast that I recommend you to check with Alpaydin's book if other ML references in your list are outdated, e.g. Cawley and Talbot.

You should also motivate why you are using a Random Forest method when there are more modern Deep Learning methods that are more powerful. (The answer is that with your limited dataset, a Random Forest method is more appropriate so you have chosen the right method - but if you had more data it would indeed be a better idea to use a neural network which can capture more nonlinear relationships.)

2: The statistical analysis is very carefully designed and builds on a selection of relevant recent work.

3: All the data (i.e. the individual answers by each test person) are made available for experimental comparison.

4: The language is indeed good overall, and the manuscript is easy to follow.

Recommendation:

However, I would recommend to let a native English speaker go over the manuscript text. There are numerous Germanic language constructions with very long sentences, that could be divided up. This would make the text easier to read. There are also some convoluted sentences that are difficult to understand, e.g.

p30: "If we were successful ... artwork evaluation."

p31: "The ability of artworks ... Wagemans, 211b)."

p34: "By employing machine ... interesting ones."

6. PLOS authors have the option to publish the peer review history of their article (what does this mean?). If published, this will include your full peer review and any attached files.

Reviewer #1: No

Reviewer #2: **Yes: **Hedvig Kjellström

---

## [Author Response · Author response to Decision Letter 0]

1 Mar 2024

Response to Editor & Reviewers

We are encouraged by the positive reception of our paper in its initial iteration and are optimistic that the revisions, guided by your feedback, have further refined and improved the manuscript.

Below, you will find our responses to each of the comments and suggestions raised. To ensure clarity and coherence, we have organized the reviewers' feedback into paragraphs, categorizing them conceptually. This is followed by our detailed reply, including specific references to the corresponding sections in the revised manuscript where in-text changes have been made.

In the revised manuscript, we have highlighted all major changes in red text for ease of identification. Additionally, we have undertaken a thorough review of the entire document, correcting language errors and typographical mistakes to enhance its readability and accuracy.

We thank you once again for your invaluable contributions. We eagerly look forward to continuing our collaboration towards the successful publication of this work.

Note, we uploaded a separate documents, Response to Reviewers, responding in detail on each of the received Reviews. 

The authors

---

## [Decision Letter · Decision Letter 1]

3 Apr 2024

PONE-D-23-31191R1Using Machine Learning to Predict Judgments on Western Visual Art Along Content-Representational and Formal-Perceptual AttributesPLOS ONE

Dear Dr. Spee,

Thank you for submitting your manuscript to PLOS ONE. After careful consideration, we feel that it has merit but does not fully meet PLOS ONE’s publication criteria as it currently stands. Therefore, we invite you to submit a revised version of the manuscript that addresses the points raised during the review process.

We look forward to receiving your revised manuscript.

Kind regards,

Maja Vukadinovic

Academic Editor

PLOS ONE

Journal Requirements:

Reviewers' comments:

Reviewer's Responses to Questions

**Comments to the Author**

1. If the authors have adequately addressed your comments raised in a previous round of review and you feel that this manuscript is now acceptable for publication, you may indicate that here to bypass the “Comments to the Author” section, enter your conflict of interest statement in the “Confidential to Editor” section, and submit your "Accept" recommendation.

Reviewer #1: (No Response)

Reviewer #2: All comments have been addressed

2. Is the manuscript technically sound, and do the data support the conclusions?

Reviewer #1: Yes

Reviewer #2: Yes

3. Has the statistical analysis been performed appropriately and rigorously? 

Reviewer #1: Yes

Reviewer #2: Yes

4. Have the authors made all data underlying the findings in their manuscript fully available?

Reviewer #1: Yes

Reviewer #2: Yes

5. Is the manuscript presented in an intelligible fashion and written in standard English?

Reviewer #1: Yes

Reviewer #2: Yes

6. Review Comments to the Author

Reviewer #1: Thank you for your detailed response to my previous comments. Regarding the unresolved issue concerning the noise ceiling, I suggest comparing your model's performance to theoretical human-level performance. This comparison is possible given that each piece of art was evaluated by multiple participants.

There are two potential methodologies for this comparison. The first involves predicting the average art judgments across participants, excluding the score from one held-out participant, using this held-out participant's annotation. This process should be repeated for each participant, thereby allowing the calculation of R^2 values for each iteration. Your model's performance can then be compared against the distribution of these R^2 values.

The second method assumes that the annotations follow a normal distribution, enabling you to calculate the mean and variance of art judgments for each piece using the participants’ annotations. With these parameters, you could simulate human annotations by sampling from the estimated normal distribution and subsequently calculate the R^2 value against the theoretical true annotation (mean of the estimated normal distribution).

This benchmarking your model against a calculated human-level performance will provide a clearer understanding of your model's relative performance.

Reviewer #2: All my comments to the previous version have been addressed in a great way. Specifically, the machine learning approach is more rigorously justified, with a relevant reference.

I am thus happy with the current version!

7. PLOS authors have the option to publish the peer review history of their article (what does this mean?). If published, this will include your full peer review and any attached files.

Reviewer #1: No

Reviewer #2: No

---

## [Author Response · Author response to Decision Letter 1]

8 May 2024

Response to Editor & Reviewers

We would like to express our gratitude to the Editor and both Reviewers for their insightful comments and reviews. We are pleased to note that Reviewer 2 is fully satisfied with the revisions we have made. Please find below our response to the comments provided by Reviewer 1.

Sincerely, 

The Authors

Reviewer 1: Thank you for your detailed response to my previous comments. Regarding the unresolved issue concerning the noise ceiling, I suggest comparing your model's performance to theoretical human-level performance. This comparison is possible given that each piece of art was evaluated by multiple participants.

There are two potential methodologies for this comparison. The first involves predicting the average art judgments across participants, excluding the score from one held-out participant, using this held-out participant's annotation. This process should be repeated for each participant, thereby allowing the calculation of R^2 values for each iteration. Your model's performance can then be compared against the distribution of these R^2 values.

The second method assumes that the annotations follow a normal distribution, enabling you to calculate the mean and variance of art judgments for each piece using the participants’ annotations. With these parameters, you could simulate human annotations by sampling from the estimated normal distribution and subsequently calculate the R^2 value against the theoretical true annotation (mean of the estimated normal distribution).

This benchmarking your model against a calculated human-level performance will provide a clearer understanding of your model's relative performance.

Reply: Thank you very much, we really appreciate your effort! 

On the one hand, we would like to point out that in the paper we already present a comparison with human performance in terms of the R² values of the art judgement predictions (Table 3 in the manuscript).

The R² values quantify the errors between models and humans (which are the squared differences between single predictions of our models and the corresponding human ratings) normalized to the amount of variability of the human ratings (which is the squared difference between single human ratings and the average of human ratings) (see Figure 1).

Figure 1: Definition of the R² value. Source: https://scikit-learn.org/stable/modules/model_evaluation.html#r2-score

On the other hand, we agree that this measure does not allow to quantify the noise ceiling, hence, an upper bound of model performance given the consistency (or, inversely, measurement noise) of the target variables. Unfortunately, we were not able to derive a method to make the requested comparison from your instructions above. Therefore, we turned to a library of Lukas Snoek (https://lukas-snoek.com) at github (https://github.com/lukassnoek/noiseceiling, https://doi.org/10.5281/zenodo.7233867) to make the computations. This library computes the maximum possible R² score given the variance in the dependent variable. For the results see Table 1 (for Table 1 see pdf version "Response to Reviewers 2"). 

For most of the art judgments our GBDT models performed better than the theoretical noise ceiling limits. However, this surprising result can be explained by the lack of true repeated measures in our data. The repeated measures used here come from different subjects. Thus, although each subject rated the same picture with respect to a particular art-judgement, each subject also rated the art-attributes of the picture accordingly. Hence, our GBDT models were able to capture relationships between art-attributes and art-judgments (as intended) and to surpass the computed theoretical limits, because parts of the art-judgement variability can be explained by the art-attributes. Hence this noise ceiling approach is not well suited in our specific case. It would be well suited if the same subjects would have made repeated judgements.

Therefore, we don’t see how this comparison is informative for readers and we wouldn’t like to include it in the paper. However, we remain open to any suggestions on this matter.

---

## [Decision Letter · Decision Letter 2]

9 May 2024

Using Machine Learning to Predict Judgments on Western Visual Art Along Content-Representational and Formal-Perceptual Attributes

PONE-D-23-31191R2

Dear Dr. Spee,

We’re pleased to inform you that your manuscript has been judged scientifically suitable for publication and will be formally accepted for publication once it meets all outstanding technical requirements.

Kind regards,

Maja Vukadinovic

Academic Editor

PLOS ONE

Reviewers' comments:

Reviewer's Responses to Questions

**Comments to the Author**

1. If the authors have adequately addressed your comments raised in a previous round of review and you feel that this manuscript is now acceptable for publication, you may indicate that here to bypass the “Comments to the Author” section, enter your conflict of interest statement in the “Confidential to Editor” section, and submit your "Accept" recommendation.

Reviewer #1: All comments have been addressed

2. Is the manuscript technically sound, and do the data support the conclusions?

Reviewer #1: Yes

3. Has the statistical analysis been performed appropriately and rigorously? 

Reviewer #1: Yes

4. Have the authors made all data underlying the findings in their manuscript fully available?

Reviewer #1: Yes

5. Is the manuscript presented in an intelligible fashion and written in standard English?

Reviewer #1: Yes

6. Review Comments to the Author

Reviewer #1: Thank you very much for your response and efforts in resolving my concern. I wanted to understand the theoretical upper limit of human participants in predicting the average human annotations and to gauge how much the model has progressed toward reaching human-level performance. However, given the limitations of the data, I agree with the authors' points.

7. PLOS authors have the option to publish the peer review history of their article (what does this mean?). If published, this will include your full peer review and any attached files.

Reviewer #1: No

---

## [Editor Report · Acceptance letter]

4 Jun 2024

PONE-D-23-31191R2 

PLOS ONE

Dear Dr. Spee, 

I'm pleased to inform you that your manuscript has been deemed suitable for publication in PLOS ONE. Congratulations! Your manuscript is now being handed over to our production team.

Kind regards, 

on behalf of

Dr. Maja Vukadinovic 

Academic Editor

PLOS ONE